# Combining the Antenatal Risk Questionnaire and the Edinburgh Postnatal Depression Scale in Early Pregnancy in Danish Antenatal Care—A Qualitative Descriptive Study

**DOI:** 10.3390/ijerph21040454

**Published:** 2024-04-08

**Authors:** Lotte Broberg, Jane M. Bendix, Katrine Røhder, Ellen Løkkegaard, Mette Væver, Julie C. Grew, Helle Johnsen, Mette Juhl, Vibeke de Lichtenberg, Michaela Schiøtz

**Affiliations:** 1Center for Clinical Research and Prevention, Bispebjerg-Frederiksberg Hospital, Nordre Fasanvej 57, 2000 Frederiksberg, Denmark; julie.grew@regionh.dk (J.C.G.); michaela.louise.schioetz@regionh.dk (M.S.); 2Department of Gynecology and Obstetrics, Slagelse Hospital, Fælledvej 14, 4200 Slagelse, Denmark; 3Department of Gynecology and Obstetrics, Copenhagen University Hospital—North Zealand, Dyrehavevej 29, 3400 Hillerød, Denmark; jane.bendix@regionh.dk (J.M.B.); ellen.christine.leth.loekkegaard@regionh.dk (E.L.); 4Center for Early Intervention and Family Research, Department of Psychology, University of Copenhagen, Øster Farimagsgade 2A, 1353 Copenhagen, Denmark; katrine.rohder@psy.ku.dk (K.R.); mette.vaever@psy.ku.dk (M.V.); 5Department of Midwifery and Therapeutic Science, University College Copenhagen, Sigurdsgade 26, 2200 Copenhagen, Denmark; hejo@kp.dk (H.J.); meju@kp.dk (M.J.); vide@kp.dk (V.d.L.)

**Keywords:** mental health, depression, pregnancy, interview

## Abstract

Pregnant women with a history of mental disorders, neglect, or low social support are at increased risk of mental health problems. It is crucial to identify psychosocial risk factors in early pregnancy to reduce the risk of short- and long-term health consequences for mother and child. The Antenatal Risk Questionnaire has been found acceptable as a psychosocial screening tool among pregnant women in Australia, but it has not been tested in a Scandinavian context. The aim of this study was to explore the experiences of pregnant women when using the Antenatal Risk Questionnaire and the Edinburgh Postnatal Depression Scale as part of a model to identify psychosocial vulnerabilities in pregnancy in Denmark. We conducted individual interviews (*n* = 18) and used thematic analysis. We identified two main themes: (1) Feeling heard and (2) An occasion for self-reflection. Overall, the pregnant women deemed the online ANRQ/EPDS acceptable as a screening tool. The screening model provided a feeling of being heard and provided an occasion for self-reflection about mental health challenges related to pregnancy and motherhood. However, some women expressed that the screening raised concerns and fear of the consequences of answering honestly. A non-judgmental, open, emphatic, and reassuring approach by clinicians may help reduce stigma.

## 1. Introduction

Maternal mental health problems are considered an increasing, major global public health challenge by the World Health Organization [1]. A systematic review found that the prevalence of perinatal depression is 12% [2], while the prevalence for a clinical diagnosis of any anxiety disorder in relation to pregnancy is 15.2% [3]. The transition to motherhood is a time of increased vulnerability to the onset or relapse of a mental disorder [4], and women with a history of mental disorders, stressful life events, abuse, or low social support are at increased risk of mental health problems such as depression and anxiety during the perinatal period [5,6]. Untreated mental health problems during pregnancy are associated with concerns regarding the potential for later developmental problems in offspring, such as attention deficit hyperactivity disorder (ADHD), depression, and anxiety, and can also affect the mother’s parenting skills [7,8,9,10,11]. Further, the mother’s well-being during pregnancy can affect her emotional attachment to the fetus, which is a critical factor for developing good parenting skills [12].

Therefore, to reduce the risk of short- and long-term consequences for mother and child, it is crucial to identify psychosocial risk factors early in pregnancy. A recently published scoping review reveals that less than half of the European countries have policies specifically targeting perinatal mental health. Moreover, only a small fraction of European countries provide screening for perinatal mental health and tailored antenatal care for this group of pregnant women [13]. The Danish Health Authority emphasizes that systematic screening of women at psychosocial risk early in pregnancy is important, but a specific tool is not recommended [14]. For several years, the EPDS has been used for screening purposes in primary care in Denmark by public health visitors and has also been validated in a Danish context for use postpartum [15]. However, the EPDS has only rarely been used in general practice or as part of the antenatal care program.

The Edinburgh Postnatal Depression Scale (EPDS) is a widely used and validated tool to detect symptoms of depression during pregnancy and postpartum [16]. However, while the EPDS assesses mood, interest or pleasure, guilt or self-blame, and other symptoms commonly associated with depression within a timeframe of seven days [17], there are psychosocial risk factors associated with adverse perinatal mental health [5,6] that the EPDS does not identify. To identify these past and present psychosocial circumstances, including perceived social support, mental health history, adverse childhood experiences, and other stressful life events, it is relevant to include a tool focusing on these factors. The Antenatal Risk Questionnaire (ANRQ), a validated self-report psychosocial assessment tool [18], has been found useful as a key element in the early identification of mental health risk and morbidity across the perinatal period [18]. Further, while the EPDS can detect current depression, it has been found that the ANRQ can predict depression six weeks after childbirth [19].

For this review, we aimed to critically analyze existing tools to measure perinatal mental health risk, recommend combining a tool that will identify current depressive or anxiety symptoms, for example, the EPDS, and a broader tool that will assess the psychosocial vulnerabilities, for example, the ANRQ [20]. In line with that, Australian Perinatal Mental Health Guidelines [21] recommend the use of the ANRQ in combination with the EPDS to identify women at increased risk of mental health problems. A survey examining pregnant Australian women’s experience with the ANRQ found that the women’s acceptability of the ANRQ was high [18,22]. However, the ANRQ has not been evaluated in an interview study and has not been tested in a Scandinavian context, which is relevant due to potential differences in the characteristics of the study population and the antenatal care program. Concerning the experience of answering the EPDS, a systematic review from 2015 including qualitative studies found that pregnant women generally found it acceptable [23]. However, a later study analyzing qualitative data from a cohort study found that some women reported difficulty due to the emotional responses triggered by the EPDS questions and the way disclosures were handled [24]. There is a lack of qualitative studies exploring pregnant women’s experience of answering the EPDS during pregnancy in a Scandinavian context.

Therefore, the present study aimed to explore the experiences of pregnant women using the ANRQ and EPDS as part of a model to identify psychosocial vulnerabilities in early pregnancy within the Danish healthcare system.

Our research questions were:How do pregnant women experience answering the ANRQ and EPDS questionnaires online?How do pregnant women perceive the follow-up procedure after completing the questionnaires?

## 2. Materials and Methods

### 2.1. Design and Setting

This was a qualitative descriptive study [25] based on semi-structured, individual interviews analyzed using reflexive thematic analysis [26]. The qualitative descriptive method is philosophically grounded in naturalistic inquiry, which values the subjective nature of data, the informant being the expert [27]. Wishing to describe the perceptions and experiences from the viewpoint of the target population as well as understand complex events embedded within the human context, the qualitative descriptive design seemed appropriate [28,29]. For analysis, we employed a reflexive approach of thematic analysis [26], which aligns with the qualitative descriptive methodology, as discussed by Vaismoradi et al. [30].

The reporting of the results follows consolidated criteria for reporting qualitative research (COREQ) [31]. The present study is nested within the First Step project, a feasibility study aiming to evaluate ANRQ/EPDS as a screening tool to identify psychosocial vulnerabilities among pregnant women and thereby strengthen the referral process in antenatal care. Pregnant women planning to attend antenatal care at the North Zealand Hospital in the Capital Region of Denmark (NOH), accommodating 4200 deliveries annually, received written information about the First Step project when they booked their first-trimester ultrasound scan. When the women arrived for the first-trimester scan, research assistants provided oral information about the project and participants were invited to the study. The women were eligible to participate if they could complete the questionnaire in Danish or English and planned to give birth at the NOH. Women consenting to participate (*n* = 774) received a link to a short online questionnaire package consisting of the ANRQ and the EPDS in Danish or English. See Figure 1. The First Step project was conducted from November 2021 to March 2022.

All women who completed the questionnaire were subsequently contacted by phone by LB or a research assistant (Figure 1). If the initial phone consultation revealed a need or a potential need for extended care, a second follow-up phone consultation with an experienced healthcare professional from the hospital visiting team was arranged. For this follow-up consultation, 45 min was allocated. During this consultation, the following antenatal care program was tailored taking the concept of shared decision-making (SDM) [32] into account, including the women’s individual needs and preferences.

### 2.2. The Antenatal Risk Questionnaire and the Edinburgh Postnatal Depression Scale

The women’s psychosocial risk profile was measured by the ANRQ developed by Marie-Paule Austin and colleagues [18]. According to an agreement with the Australian research group that developed the ANRQ, we conducted a backward–forward translation from English to Danish using independent translators and pilot testing of the translated items. The ANRQ is a 12-item scale modified version of the 23-item Postnatal Risk Questionnaire and assesses the following domains: emotional support from the individual’s mother in childhood; history of depressed mood or mental illness and treatment received; perceived level of support available following the birth of the baby; partner emotional support; life stresses in previous 12 months; personality style (anxious or perfectionistic traits); and history of abuse. The ANRQ is a validated scale and has a sensitivity of 0.62 and a specificity of 0.64 [18]. It consists of a combination of categorical and continuous data, and responders are given the option to describe stressful events, changes, or losses during the last 12 months or other worries in general. Categorical questions are scored 0 if the answer is “no” and 5 if the answer is “yes.” Continuous questions are scored on a Likert scale of 1–5 or 1–6. In total, the 12 items yield a maximum possible score of 62 and a minimum score of 5. Symptoms of depression were measured by the EPDS developed by Cox and Davidson in 1987 [17]. The scale consists of 10 items scored on a 4-point Likert scale (0–3). The lowest score is 0 and the highest score is 30 [17]. Based on the literature, we used an ANRQ score of ≥23 to indicate high psychosocial risk [18] and an EPDS score of ≥11 to indicate risk of clinical depression, with a sensitivity of 0.81 (0.75 to 0.87) and a specificity of 0.88 (0.85 to 0.91) [16]. If the ANRQ or EPDS cut-off was met or if the woman’s remarks in the free-text field gave cause for concern, the screening result was considered high risk (Figure 1).

### 2.3. Sampling

Informants were pregnant women referred to the NOH, and to ensure that information-rich cases were recruited, informants were purposefully selected [33]. To capture a wide range of experiences and perspectives, a maximum variation sampling strategy was applied, aiming to ensure that the informants were broadly represented in terms of age, parity, ethnicity, cohabitation status, and employment situation, and whether they had been referred to extended antenatal care based on the screening.

### 2.4. Recruitment of Informants

During enrolment into the First Step project, informants consented to be contacted for further information about this nested qualitative study. Informants in the present study were contacted via email from December 2021 to January 2023. In cases of non-response to the email, up to two short text messages were made. A total of 40 pregnant women were invited to participate, out of which 18 agreed to participate, 17 did not respond, and 5 declined to participate due to lack of time.

### 2.5. Ethical Considerations

In Denmark, studies that do not involve human biological material need not be reported to an ethics committee [34]. The approval for this study was given by the Danish Data Protection Agency (P-2021-575). All informants were given oral and written information about the study and informed written consent was obtained. The informants were informed that their data would be kept confidential and anonymous.

All informants were informed that they could withdraw from the study at any time without giving any reason. Moreover, given the sensitive nature of the subject matter, a post-interview debriefing was included at the end of the interview. During this session, women’s interview experiences were explored, and they were encouraged to contact the interviewer or their midwife in case of further need to talk about the situation.

### 2.6. Data Collection

Interviews were carried out by a postdoc on the First Step project, LB (F), from December 2021 to January 2023. LB has worked as a midwife for several years. She has experience with conversations about difficult topics, both in consultations and in connection with research interviews. LB discussed assumptions in this field with other researchers from the group before the interviews to be more aware of her own assumptions and capacity to set them aside. Among the assumptions was that adverse childhood experiences may evoke strong emotions in the women, requiring a supportive and empathic approach from the healthcare professional. The research group consisted of a group of researchers from different professional and scientific backgrounds, including psychologists, midwives, an obstetrician and candidates in public health.

LB did not participate in the recruitment process for the First Step program, but conducted initial phone calls with informants after they completed questionnaires and invited them to the interviews. Interviews with pregnant women were conducted face to face in their homes (*n* = 2) or by telephone (*n* = 16) depending on the participant’s preference and considering the constraints imposed by the COVID-19 pandemic. The interviews were conducted using a semi-structured interview guide consisting of open-ended questions to ensure consistency and address the research questions [35]. The themes in the guide were focused on the women’s pregnancies, their social support, and their experiences of answering the questionnaire package consisting of the ANRQ and the EPDS. All informants were asked the same opening question: “Can you please tell me a little about yourself?” When asking about the ANRQ and the EPDS, some of the questions from the two questionnaires were given as examples, making it possible to distinguish the two different questionnaires from each other. Further, there were questions about their experience of the follow-up telephone consultation.

The interviewer encouraged informants to speak freely and elaborate on their statements, emphasizing that there were no right or wrong answers [35]. Saturation was sought by gathering as much richness of information as possible during each interview, including continuing until a full understanding of the participant’s perspective was reached and probing the informants to provide examples and elaborate on their experiences. This resulted in identified codes or themes being satisfactorily exemplified in the data [36]. The interviews yielded highly dense descriptions, and after 15 interviews, a high rate of recurring responses was seen, indicating emerging theoretical saturation [36].The interviews were digitally recorded verbatim by research assistants and ranged from 32 to 75 min in duration.

### 2.7. Data Analysis

Data were analyzed using thematic analysis [26] consisting of six phases, as follows.

Familiarizing with data: The interviews were transcribed verbatim by a research assistant and the first author (LB) read and reread the material, noting down initial ideas on meanings and patterns.Generating initial codes: Using the software program NVivo version 12, QRS international, the initial codes were generated by encoding sections or sentences of the data. To increase trustworthiness, a research assistant independently coded four of the interviews. Discrepancies in the analyses were discussed and the coding frameworks were integrated.Searching for themes: LB organized the initial codes into potential themes. The authors MS and LB discussed the themes until agreement was reached.Reviewing themes, including moving back and forth between themes, codes, and quotes. The authors discussed and revised the themes until the entire data set was covered and no themes overlapped.Defining and naming themes, including describing what each theme represented. To ensure inductive thematic saturation, potential themes were reviewed in a recursive process. LB, MS, and KR discussed codes, subthemes and themes until no new themes appeared and each theme mutually excluded other themes and saturation was achieved [36].Producing the report, including writing down the analysis.

The names provided alongside the quotes in the following section are pseudonyms.

## 3. Results

Characteristics of the participating pregnant women are presented in Table 1.

Thematic analysis identified two main themes and five subthemes (Table 2).

### 3.1. Theme 1. Feeling Heard

#### 3.1.1. Being Taken by the Hand

Overall, the informants experienced that being asked about their background and mental health in the questionnaires made them feel heard and taken by the hand, and some expressed a sense of relief in sharing thoughts and feelings. The possible answers and the free-text fields in the ANRQ enabled respondents to express troublesome thoughts or feelings, thereby fostering a more accepting environment for difficult emotions and offering a sense of reassurance:

*There was definitely a sense of relief in answering, and you could see that, okay, it’s okay to write that it’s a bit difficult—I mean, it’s probably not just me who thinks it’s difficult if it’s written here* [referring to the wording of the questions, red].(Cathrine)

Some of the informants described a feeling of isolation during the initial stages of their pregnancy, and they were unsure about whom to contact within the healthcare system. Being asked about their well-being in the questionnaires and being offered support afterwards was therefore experienced as a relief for these women. One expressed uncertainty about whether it was appropriate or relevant to raise severe mental health concerns with her GP. She was hesitant to inform others that she was pregnant because she did not feel grateful or happy, and the questionnaires provided a safe opportunity to express her difficult emotions:

*I felt comfortable* [completing the questionnaires, red], *like “Okay, I put everything right here, and that’s it.” I’m very grateful for this project because otherwise, I think I would never have talked about it* [the difficult feelings, red].(Paula)

While some women described having previously talked to a healthcare professional about their concerns, including adverse childhood experiences, several women said that they had never talked to a healthcare professional about these concerns and had never been asked before. Several described the combination of the ANRQ and EPDS as very good, as the ANRQ has a longer timeframe and covers childhood experiences. A few women described being initially surprised when being asked about their mental health and childhood experiences, and some found the questions emotionally demanding. As one explained when she saw the questions:

*I had to pull myself together again—you are used to being asked questions abouts physical health, so I had to spend some time to pull myself together again*.(Anne)

Further, the informants explained that the follow-up telephone consultation(s) provided an opportunity to talk about the pregnancy from their perspective. Many described the second phone consultation as an experience of receiving genuine interest and empathic care:

*She was like the first person since the pregnancy last year* [“a traumatic experience,” red] *who really took the time and spoke to me in a very caring tone. And yes, she just really listened and respected our … that we also found it difficult, and it was just really, really nice and important to me. So, I thought it was a really good opportunity for us to kind of get rid of those feelings because I don’t think there has been an opportunity to do that. I mean, how we really feel. There hasn’t really been anyone who asked us*.(Fiona)

Overall, the informants described the telephone consultation during which their scores on ANRQ/EPDS were followed up on as providing a comprehensive assessment of individual needs, offering information about relevant initiatives in antenatal care as well as making them feel welcome. This left the informants feeling reassured, confident, and better prepared for the remainder of their pregnancy. Some described feeling worried and having investigated the possibility of extended care before the phone consultation, but afterwards, they no longer felt the need:

*Yes, because it gave me some peace of mind that there are people who know how I feel. And it was said that—”Okay, it’s normal, and if it doesn’t get better, then you do this and this.” Okay, so then I knew how to deal with it*.(Cathrine)

#### 3.1.2. Creating Space for the Mental Aspects of Being Pregnant

Many informants described that standard antenatal care primarily focused on the physical aspects of pregnancy, and some of them described how the healthcare professionals’ focus on the physical aspects of pregnancy acts as a barrier to discussing mental health issues, worries, or thoughts:

*It is all about urine, umbilical cords, amniotic fluid, and placentas, and you can feel a bit overlooked if you’re struggling with some emotions*.(Anne)

However, receiving the questionnaire made the informants feel that the healthcare system was concerned about their mental well-being, which they described as a positive experience:


*I was surprised by what it was about—it was actually about how I am feeling! Inside my head. It was actually nice that—that someone wanted to know how I’m doing.*
(Maria)

In general, the informants highlighted that focusing on mental well-being, including the normal psychological processes related to the transition to motherhood, is important to them. Several informants expressed that openness towards mental health issues would be helpful for them and would promote a feeling of not being alone with difficult thoughts and feelings.

### 3.2. Theme 2. An Occasion for Self-Reflection

#### 3.2.1. Becoming Aware of the Need for Extended Care

For some, the screening process created an awareness that they needed more support and care than first realized. One informant described how she felt talking to the healthcare professional after fulfilling the questionnaire:

*Well, at that time I got a little scared because I hadn’t realized it* [being severely depressive, red] *myself*.”(Beatrice)

She described being grateful for healthcare professionals asking, taking responsibility, and helping her. For some informants, it was a challenge and a stepwise process to realize that they needed extended care:

*Yes, extended care sounds…well, at first I thought: “But there is nothing wrong with us, really!” But it’s all about accepting the help available. So it…it felt really, really good and we were both very happy afterwards*.(Louise)

Some informants expressed concern about what the healthcare professionals would think about them in relation to their answers in the questionnaires. One informant described how she disagreed with the healthcare professionals’ assessment of the need for extended care:

*Well, I think I was actually a little shaken about the fact that she thought it would be a good idea to offer it* [additional care, red] *to me.”*(Anne)

Some multiparous informants described reflecting on whether their perception of not needing extended care was accurate. As Ellie said:

“*Being offered extended care made me think: God, perhaps it might have been a good idea* [to get extended care, red] *after all?”*

Despite their initial confidence in their mothering abilities, the screening raised many thoughts and concerns about being a mother and for a period led them to doubt their abilities.

In contrast, other informants said that by telling their stories, they became more aware of what they had overcome and of their strength and ability to cope with even very adverse experiences. As Daniella, who was referred to extended care, described it:

*We talked about some of the things* [being stressed and a former traumatic birth experience, red] *and that I had handled things in the only right way one could!*

Even those who declined the extended antenatal care because they did not feel the need for it still regarded the screening as a valuable process.

#### 3.2.2. Doubts about Giving Honest Answers

Some women described that they were not giving honest answers in the questionnaire, because they were not interested in receiving extended care. Fear of prejudice about needing extended care during pregnancy, of being stigmatized as a mother who cannot take care of her child, and of the potential consequences influenced some women’s answers to the questionnaire:

*Sometimes I feel like I have to be incredibly strong now because I am going to be a mother. And sometimes one can be inclined to decline things like that* [answering the questionnaires] *because one doesn’t want to be labelled, and it can also be a bit risky when it’s professionals* [who receive the information, red].(Anne)

One informant, who was expecting her second child, said that she did not dare to submit the questionnaire because she was afraid that healthcare professionals would think that she could not handle being a mother due to her history of mental health and social issues:

*I remember thinking like: “No, I’ll cancel this questionnaire, I’m not going to participate after all, and then I think I pressed the wrong button and sent it anyway”* [laughs].(Liva)

In addition to their desire to avoid stigma and concerns about potential consequences of the healthcare professionals identifying them as having psychosocial problems, some women also thought about whether they felt ready to accept extended care and what it would entail. Some informants also considered whether they were interested in extended support when answering the questionnaire:


*When I had to answer the questionnaires, I actually sat there and thought, you know, pondered a bit about “What should I answer here?” because sometimes you choose to answer honestly, and sometimes you choose to answer based on “Do I really feel like I need help right now, if it were offered to me?”*
(Josephine)

#### 3.2.3. Perceptions of Answering the Questionnaires Online

Overall, the possibility of answering online was perceived as acceptable and advantageous. The informants explained that because the questionnaire was online, they perceived it as an unintimidating approach that gave them time and privacy to complete when the time felt right. Some described how they needed to reflect before filling out the questionnaire, and that it gave rise to a dialogue with their partner about their mental well-being and thoughts about the pregnancy. One informant described it like this:

*It mattered that it* [the questionnaire, red] *was electronic because there was more time to respond. I was honest in a way that I wouldn’t have been if I were with the midwife. I haven’t told anyone about the abuse* [sexual abuse as a child, red] *before, only my family. It stirred up old experiences to receive the questionnaire, but it was OK. It meant a lot to be able to sit at home and have time to answer the questionnaire calmly, with time for reflection. I didn’t have to answer immediately and I could talk to my husband about the questions. I didn’t go into my upbringing with the midwife—it was more like food and things like that we talked about*.(Mary)

In contrast, some informants highlighted the advantages of an in-person approach, and one argued that continuity would be strengthened if the screening were conducted by one’s own midwife. In general, the women said it was appropriate to receive the questionnaire early in pregnancy. One described how she found it undesirable that the healthcare professional who called her was not the midwife she met at the first consultation. Some reflected on the short timeframe of seven days in the EPDS and said that it was nice to have an opportunity to explain that their high EPDS score reflected intensive nausea, being tired, etc.

## 4. Discussion

This study aimed to explore the experiences of pregnant women using the ANRQ and EPDS as part of a model to identify psychosocial vulnerabilities in early pregnancy within the Danish healthcare system. Thematic analysis identified the two main themes of “Feeling heard” and “An occasion for self-reflection” and five subthemes.

A main finding was that the process of answering the online ANRQ and EPDS provided the informants an experience of feeling heard and a feeling of relief. This is in line with another study, though not related to screening, that found a positive association between online sharing of emotions and the alleviation of negative emotions [37]. The informants experienced the follow-up phone consultation with a healthcare professional after completion of the questionnaires as important, reinforcing a feeling of being heard and providing a sense of involvement in the antenatal care plan [38]. Through being seen as unique individuals, having sufficient time, and making shared decisions about their care, the informants experienced deliberate attention to their personal needs from health professionals who carefully considered the values and priorities of each woman and her unique family. In the present study, we took into account key elements from the concept of SDM, such as taking the patient’s needs and preferences into account, describing options for care, and supporting deliberation [32]. A systematic review found that SDM is significantly positively associated with outcomes such as satisfaction with care, stress management, and general health [39]. Consistently with other research [40], our results indicate that it might be relevant to work more consciously and purposefully with the principles of SDM when planning antenatal care.

The women in our study described that the follow-up consultation provided them with a feeling of being listened to and supported in a compassionate and empathic way, which prompted a feeling of being reassured and confident. As a result of this, some informants no longer felt the need for extended care. This is in line with other research, showing that warm and empathic communication between pregnant women and antenatal providers can normalize the pregnant women’s fears, reduce anxiety, and increase positive expectations of treatment [41], as well as allow the women to understand information more easily and apply it [42]. Screening for adverse psychosocial experiences and afterwards being offered the opportunity to talk about their given replies and feeling listened to has been described as a therapeutic intervention in itself, and a way to reduce traumatic shame and the stress-related symptoms shame can cause [43]. Empathic communication is a fundamental component of woman-centered midwifery care [44], and our study shows that woman-centered midwifery can play an important role already in the referral process during early pregnancy.

In general, our findings suggest that answering the ANRQ and EPDS and the follow-up procedure was deemed acceptable by the informants, which corresponds with previous studies of screening with the ANRQ [18] and the EPDS [23,45]. Further, answering the ANRQ and EPDS was perceived as an opportunity for self-reflection regarding their own mental health. The open-ended questions and free-text fields in the questionnaire were appreciated by the informants, which is consistent with the results from a study concluding that closed-ended questions can make the responder feel limited and miss the opportunity to have an open discussion about their mental health [46]. Many informants said that they had never been asked questions about adverse childhood experiences before, which is in line with other research [47], but said that they found the screening process valuable. However, some women said that the questionnaires also raised thoughts and concerns, which is in line with another qualitative study’s finding that answering the EPDS triggered emotional responses [23]. We also found that some expressed having doubts about giving honest answers. This doubt was expressed as rooted in fear of being stigmatized and being seen as a vulnerable mother and the possible negative consequences. Despite an increased public awareness of mental health in recent years, fear of stigmatization, self-stigma, and of negative perceptions by healthcare providers are barriers to sharing mental vulnerabilities previously documented in the literature [46,48,49]. Further, it is well known that uncertainty about the consequences of disclosure is a commonly identified barrier to revealing sensitive information [50,51]. In Denmark, healthcare professionals are required to refer to the authorities if concerned about a child’s well-being [52], and a recent Danish study found that encounters with healthcare providers can be experienced as being associated with some risk [53]. Therefore, it is crucial for healthcare professionals to reflect how pregnant women perceive their role and to demonstrate a critical understanding of their own position [54]. Recognizing the dual nature of support and surveillance may help explain why some pregnant women might be apprehensive about revealing sensitive information and accepting services, given that healthcare providers also wield authority over them.

Some of the informants who declined the offer of extended care described how being offered special care raised many thoughts and concerns about being a mother. This is consistent with findings from a previous Danish study from 2018. This study showed that being offered an intervention targeting vulnerable women may induce negative feelings in relation to stigmatization and self-doubt about their ability to cope with motherhood [55]. A non-judgmental, open, and reassuring approach from clinicians may help to reduce stigma and fears, contributing with honest responses and improving early diagnosis and treatment of mental health problems [50].

Overall, our informants appreciated being asked about their mental well-being, as they described how healthcare professionals’ primary focus is on the physical aspects of pregnancy, which acts as a barrier to discussing personal mental health issues and concerns. Research has shown that women may feel disempowered if pregnancy is framed as a highly medicalized process with a strong focus on interventions and monitoring [56]. Therefore, it is important that healthcare professionals adopt a holistic approach that considers not only physical health but also the mental health perspective, social circumstances, and personal concerns. By addressing these aspects throughout pregnancy, healthcare professionals can provide more comprehensive and personalized antenatal care [53].

The informants found answering the questionnaires online to be acceptable and perceived it as an unintimidating approach that gave them time and privacy to complete the questionnaires. These results align with previous findings reporting that limited time and feeling rushed are common reasons for not completing a psychosocial assessment [46] and that privacy is one of the most important concerns for pregnant women when deciding whether to disclose adverse childhood experiences [57]. In contrast, some informants highlighted the possible advantages of an in-person approach and that continuity in care would be strengthened if the screening were conducted by one’s own midwife. These findings suggest that there are different needs and preferences when screening for social and mental health issues. This is in alignment with other studies finding that women with known mental health issues prefer an in-person approach, while women without a known mental health history but nevertheless struggle with emotional problems prefer less interactive approaches and reported reluctance to share their concerns [47,50,58]. However, in some antenatal clinics, there might be a conflict between screening early in pregnancy and the possibility of being screened by one’s midwife due to structural reasons. Further, receiving appropriate antenatal care from the beginning of the pregnancy is not only a way to increase continuity of antenatal care but also to optimize cost-effectiveness.

### Main Strengths and Limitations

We employed a maximum variation sampling strategy, thereby allowing a diverse group of women to share their experiences. However, only 18 of the invited 40 women wished to participate, and the did does not include skilled workers or the non-educated, which is a limitation. Further, most informants were of Danish origin, and the cultural acceptability of being asked about mental health and social circumstances may differ between countries. Employing a maximum variation strategy provided us with valuable insights and perspectives, and the diversity contributed to uncovering a broad range of experiences. However, while only a few informants had low socioeconomic status and/or background other than Danish, there might be aspects and nuances we did not uncover.

Although we invited women who did not answer the questionnaire, unfortunately, we were not able to include any informants from this non-reply group. However, we received valuable information from one participant who submitted the questionnaire by mistake despite her intentions.

The first author, LB, conducted the qualitative interviews with the participating pregnant women. Being interviewed by a researcher from the project may have incurred a social desirability bias that may have influenced the informant’s description of their experience. However, we have reason to believe that the interviews give a trustworthy picture of the women’s experiences related to answering the questionnaire, as satisfaction and acceptability was pronounced across the interviews. The study was conducted at an antenatal clinic in a hospital in Denmark, and generalizability to other healthcare systems may be limited.

The study was strengthened by the steps taken to ensure trustworthiness, including member checking on the spot, a detailed description of the methodology used, and employing a second coder to enhance dependability and thereby trustworthiness [35]. Additionally, the diverse professional and scientific backgrounds of the research team offered multiple perspectives and reflexivity, reducing potential investigator bias and increasing confirmability [35].

## 5. Conclusions

Overall, the pregnant women experienced the use of the online ANRQ/EPDS as a screening tool to be acceptable and valuable. According to the women, the screening model provided a feeling of being heard, a room for self-reflection, and an important space for the mental aspects of being pregnant. In addition to the online questionnaire itself, the follow-up and the relational aspects therein were also described as significant for the women, which is noteworthy and a finding of clinical importance. However, caregivers must be aware that some women expressed fear of the consequences of answering honestly. A non-judgmental, open, empathic, and reassuring approach by clinicians may help to reduce stigma and promote an antenatal care codex that provides safe spaces for pregnant women to discuss mental health concerns with their healthcare provider. The present study was conducted in a high-income Western country. Future research should focus on examining the feasibility and experience of the screening model in low-income countries.

## Figures and Tables

**Figure 1 ijerph-21-00454-f001:**
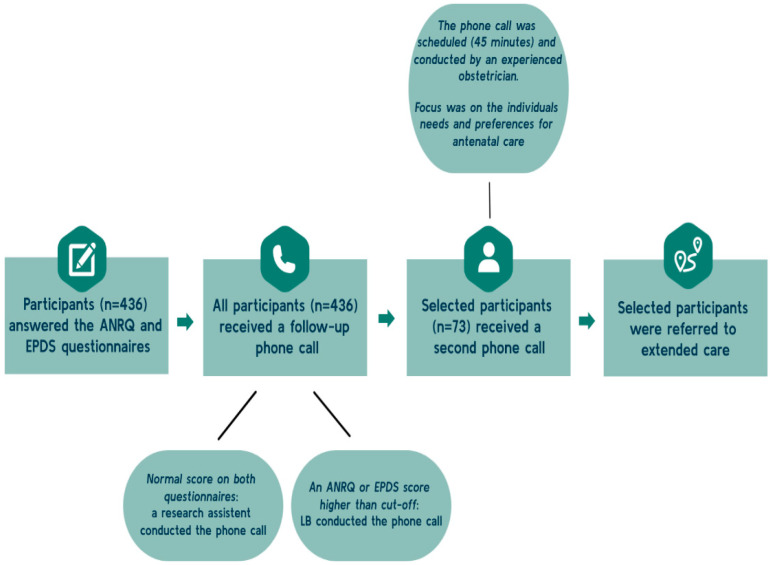
Overview of the screening procedure. ANRQ: Antenatal Risk Questionnaire. EPDS: Edinburgh Postnatal Depression Scale.

**Table 1 ijerph-21-00454-t001:** Maternal characteristics.

Maternal Characteristics	
**Age at interview years, mean (range)**	33 (29–36)
**Gestational** week, median (range)	17 + 5 (11 + 5–29 + 4)
**Parity** *n* (%) Nulliparous Parous	4 (22) 14 (78)
**Ethnicity** *n* (%) Danish ethnic origin Other ethnic origins	16 (88) 2 (12)
**Cohabitant** *n* (%) Yes No	18 (100) 0 (0)
**Education level** *n* (%) Academic 3–4 years of education (after secondary schools +/− high schools) 1–2 years of education (after secondary schools +/− high schools) Skilled worker No education	7 (39) 9 (50) 2 (11) 0 (0) 0 (0)
**Employment status** *n* (%) Employed Unemployed Student	17 (94) 1 (6) 0 (0)
**Referral (screening)** *n* (%) Offered extended antenatal care, accepted Offered extended antenatal care, declined Not offered extended antenatal care	4 (22) 2 (11) 12 (67)

**Table 2 ijerph-21-00454-t002:** An overview of main themes and subthemes.

Main Themes	Subthemes
**Feeling heard**	Being taken by the hand Creating space for the mental aspects of being pregnant
**An occasion for self-reflection**	Becoming aware of the need for extended care Doubts about giving honest answers Advantages of answering the questionnaires online

## Data Availability

According to general data protection regulations, the qualitative data are confidential and cannot be provided.

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
