# Peer review of "Combining the Antenatal Risk Questionnaire and the Edinburgh Postnatal Depression Scale in Early Pregnancy in Danish Antenatal Care—A Qualitative Descriptive Study"

_ijerph, 2024, doi:10.3390/ijerph21040454_

Round 1

Reviewer 1 Report

Comments and Suggestions for Authors

Thank you for the opportunity to review the manuscript entitled “Experiences with the Antenatal Risk Questionnaire in combination with the Edinburgh Postnatal Depression Scale in early pregnancy in Danish antenatal care - a qualitative descriptive  study”. This is a relevant and important topic. I appreciate the aim, the importance given to qualitative research in this field, the figures provided, and the contents of this paper. My comments are to help add clarity to the paper:

1)        In the Introduction, the authors reported the WHO report of 2009. In recent years, due to the COVID-19 spread, the prevalence of anxiety and depression has largely increased. I suggest report update GLOBAL data on this phenomenon, such as the global prevalence of depression and anxiety in the perinatal period, and WHO recent guidelines (WHO, 2023). Considering that the authors in the introduction compared a European country (Denmark) with Australia, some information on perinatal maternal mental health and healthcare guidelines in Europe could help to reinforce the importance of the paper for the scientific community, giving a broader view beyond the Denmark territory.

2)        After the reported quotes, women’s names are given. Are they pseudonyms? This should be clarified in the methodology.

3)        To ensure transparency of the procedure, a script of the complete interview should be provided.

4)        More information on saturation should be provided. In particular, how did the authors deal with the heterogeneity of participants? Saturation depends, in fact, on the homogeneity and heterogeneity of the study populations. Few women in the group of participants have for example low level of education, or are no cohabitant (0), or other ethics groups, or nulliparous. Being these elements at-risk factors for maternal mental health, how did it affect saturation?

5)        In methods, please give more space to the description of the research team. An explicit description of the researchers’ background, assumptions, and how the authors protected themselves against bias and/or emotionality is fundamental in qualitative research.

6)        I think referring to interviews in terms of “data” is not completely in line with the objective and the characteristics of qualitative research.

7)        I suggest inserting participants’ quotes to support these two parts of the results: 1) “Some multiparous informants described reflecting on whether their perception of not needing extended care was accurate …” 2) “other informants said that by telling their stories they became more aware of what they had overcome and of their strength and ability to cope with even very adverse experiences”

8)        A lot of results refer directly or indirectly to relational aspects related to who entered into contact with the women, who did the screening, and so on. I think you should flag this aspect because to date most of the screening for perinatal depression or anxiety is done online without the direct involvement of a professional and in particular of a TRUSTED professional who could support and hold the possible emotions lived by women answering the questions. What clinical recommendations does the paper offer from its findings, and are they applicable to other contexts?

9)        Abbreviations in tables, figures, and text need to be clarified. Please, be consistent with the format in which the author decided to report the pseudonyms of the women (italic or capital).

Author Response

Reviewer 1

Thank you for the opportunity to review the manuscript entitled “Experiences with the Antenatal Risk Questionnaire in combination with the Edinburgh Postnatal Depression Scale in early pregnancy in Danish antenatal care - a qualitative descriptive study”. This is a relevant and important topic. I appreciate the aim, the importance given to qualitative research in this field, the figures provided, and the contents of this paper. My comments are to help add clarity to the paper:

Comment 1a:

In the Introduction, the authors reported the WHO report of 2009. In recent years, due to the COVID-19 spread, the prevalence of anxiety and depression has largely increased. I suggest report update GLOBAL data on this phenomenon, such as the global prevalence of depression and anxiety in the perinatal period, and WHO recent guidelines (WHO, 2023).

The author's response to comment 1a:

Thank you for this comment. We have added the recent WHO guidelines (WHO, 2023) and added the following sentence in line 38-40:

“A systematic review found that the prevalence of perinatal depression is 12% (ref) while the overall prevalence for a clinical diagnosis of any anxiety disorder in relation to pregnancy was 15.2% (ref)”.

According to perinatal depression, we refer to the same systematic review (Woody et al) as the WHO report from 2023.

Comment 1b:

Considering that the authors in the introduction compared a European country (Denmark) with Australia, some information on perinatal maternal mental health and healthcare guidelines in Europe could help to reinforce the importance of the paper for the scientific community, giving a broader view beyond the Denmark territory.

The author's response to comment 1b:

Thank you for this important comment.

We have added the following paragraph in the Introduction, line 51-55:

“A recently published scoping review reveals that less than half of the European countries have policies specifically targeting perinatal mental health. Moreover, only a small fraction of European countries provide screening for perinatal mental health and tailored antenatal care for this group of pregnant women”.

Comment 2:

After the reported quotes, women’s names are given. Are they pseudonyms? This should be clarified in the methodology.

The author's response to comment 2:

Thank you for this comment. The names are pseudonyms and we have added the following sentence at the end of the method section, line 259:

“The names provided after the quotes in the following section are pseudonyms”.

Comment 3:

To ensure transparency of the procedure, a script of the complete interview should be provided.

The author's response to comment 3:

According to the General Data Protection Regulation, the qualitative data are confidential and unfortunately cannot be provided.

Comment 4:

More information on saturation should be provided. In particular, how did the authors deal with the heterogeneity of participants? Saturation depends, in fact, on the homogeneity and heterogeneity of the study populations. Few women in the group of participants have for example low level of education, or are no cohabitant (0), or other ethics groups, or nulliparous. Being these elements at-risk factors for maternal mental health, how did it affect saturation?

Authors' response to comment 4:

Thank you for this important comment.

We have added the following sentence in the Data collection section, line 232-234:

“The interviews yielded highly dense descriptions and after 15 interviews a high rate of recurring responses were seen, indicating emerging theoretical saturation (Saunders)”

And in the section on strengths and limitations, we added the following sentence, line 523-527:

“Employing a maximum variation strategy provided us with valuable insights and perspectives, and the diversity contributed to uncovering a broad range of experiences. However, while only a few informants had low socioeconomic status and/or background other than Danish, there might be aspects and nuances we did not uncover”

Comment 5:

In methods, please give more space to the description of the research team. An explicit description of the researchers’ background, assumptions, and how the authors protected themselves against bias and/or emotionality is fundamental in qualitative research.

Authors' response to comment 5:

Thank you for this comment. We agree and have added the following in the method section, line

200-210:

“Interviews were carried out by postdoc on ‘The First Step’ LB (F) from December 2021 to January 2023. LB has worked as a midwife for several years. She has experience with conversations about difficult topics, both in consultations and in connection with research interviews. LB discussed assumptions in this field with other researchers from the group before the interviews, to be more aware of own assumptions and capable of setting them aside. Among assumptions was, that adverse childhood experiences may evoke strong emotions in the women, requiring a supportive and empathic approach from the healthcare professional. The research group consisted of a group of researchers from different professional and scientific backgrounds, including psychologists, midwives, a doctor and candidates in Public Health.

Comment 6:

I think referring to interviews in terms of “data” is not completely in line with the objective and the characteristics of qualitative research.

Authors' response to comment 6:

Thank you. Since our sources (Sandelowski and Braun/Clark) use the term “data” about interviews, we have chosen to use this word as well.

Comment 7:

I suggest inserting participants’ quotes to support these two parts of the results: 1) “Some multiparous informants described reflecting on whether their perception of not needing extended care was accurate …” 2) “other informants said that by telling their stories they became more aware of what they had overcome and of their strength and ability to cope with even very adverse experiences”

Authors' response to comment 7:

Thank you for this comment. We agree and have added the following quotes, line 355-357:

”Being offered extended care made me think: God, perhaps it might have been a good idea (to get extended care, red.) after all?

And in line 363-365:

“We talked about some of the things (being stressed because of work and a former traumatic birth experience, red.), and that I had handled things in the only right way one could

Comment 8:

A lot of results refer directly or indirectly to relational aspects related to who entered into contact with the women, who did the screening, and so on. I think you should flag this aspect because to date most of the screening for perinatal depression or anxiety is done online without the direct involvement of a professional and in particular of a TRUSTED professional who could support and hold the possible emotions lived by women answering the questions. What clinical recommendations does the paper offer from its findings, and are they applicable to other contexts?

Authors' response to comment 8:

Thank you. We added the following sentence in the conclusion, line 551-553:

“In addition to the online questionnaire itself, the follow-up and the relational aspect therein were also described as significant for the women which is noteworthy and a finding of clinical importance”

Comment 9:

Abbreviations in tables, figures, and text need to be clarified. Please, be consistent with the format in which the author decided to report the pseudonyms of the women (italic or capital).

Authors' response to comment 9:

Thank you. We have clarified abbreviations in the tables, figures, and text. Further, we now use the same (capital) format reporting the pseudonyms of the women.

Reviewer 2 Report

Comments and Suggestions for Authors 1. Please retitle. Is it too long?

2. Please check your keyword with MeshBrowser

3. In Line 53 the author should find out EPDS use in the context of your country

4. Line 101 Did you name your project? you should use it instead of ANRQ/EPDS. 

5. Line 122 Is it needed to review the whole antenatal care in Denmark? It is general information. You could remove the section.  But, the ANRQ and EPDS review is needed.

6. Line 199, The details of the question are not necessary.

7. Lines 443-450 could be removed, you should imply what you need to do with the project from the results of this study.

8. Line 461-478 These data should be included in the results. 

9. You should suggest your results for other countries, especially in LMIC could apply PHQ-9 as it is the regular screening for perinatal depression, such as India (R. Ransing, S.N. Deshpande, S.R. Shete, I. Patil, P. Kukreti, P. Raghuveer, et al.Assessing antenatal depression in primary care with the PHQ-2 and PHQ-9: can it be carried out by auxiliary nurse midwife (ANM)? Asian Journal of Psychiatry, 53 (2020), Article 102109) and Thailand. (Sawaddisan, R., Ransing, R., & Jatchavala, C. (2023). Concordance of the Thai versions of the Patient Health Questionnaire and Edinburgh Post-natal Depression Scale for antenatal depression. Journal of Health Science and Medical Research, 41(6), e2023985.), or recommend other screening tools (Srisurapanont, M.; Oon-arom, A.; Suradom, C.; Luewan, S.; Kawilapat, S. Convergent Validity of the Edinburgh Postnatal Depression Scale and the Patient Health Questionnaire (PHQ-9) in Pregnant and Postpartum Women: Their Construct Correlations with Functional Disability. Healthcare 2023, 11, 699.).

10. up-date the reference, it should be 2014-2024. 

Author Response

Reviewer 2

Comment 1:

Please retitle. Is it too long?

Authors' response to comment 1:

Thank you. We agree and have shortened the title from:

“Experiences with the Antenatal Risk Questionnaire in combination with the Edinburgh Postnatal Depression Scale in early pregnancy in Danish antenatal care - a qualitative descriptive study”

To:

“Combining the Antenatal Risk Questionnaire and the Edinburgh Postnatal Depression Scale in early pregnancy in Danish antenatal care - a qualitative descriptive study”

Comment 2:

Please check your keyword with MeshBrowser

Authors' response to comment 2:

Thank you. We have changed the keywords, so we only use MeSH terms now

Comment 3:

 In Line 53 the author should find out EPDS use in the context of your country

Authors' response to comment 3:

Thank you for this comment, in line 57-60 we have added the sentence:

“For several years, the EPDS has been used for screening purposes in primary care in Denmark by public health visitors and it has also been validated in a Danish context for use postpartum [4]. However, the EPDS has only rarely been used in general practice or as part of the antenatal care program.

Comment 4: Line 101 Did you name your project? you should use it instead of ANRQ/EPDS. 

Authors' response to comment 4:

Thank you. The project “The First Step” encompasses both a quantitative and the present qualitative study. We have chosen to name the questionnaire “ANRQ/EPDS” so that it is clear to the reader what we are investigating.

Comment 5:

Line 122 Is it needed to review the whole antenatal care in Denmark? It is general information. You could remove the section.  But, the ANRQ and EPDS review is needed.

Authors' response to comment 5:

Thank you for this comment. We agree and have removed the section Antenatal care in Denmark and kept the description of the ANRQ and the EPDS.

Comment 6:

Line 199, The details of the question are not necessary.

Authors' response to comment 6:

Thank you for this remark. We have changed the text from:

“All informants were asked the same opening question: “Can you please tell me a little about yourself?”. Some of the following questions were: “Can you tell me about your considerations when you answered the questionnaire?” and “Did any of the questions, in particular, make an impression?”.”

To:

All informants were asked the same opening question: “Can you please tell me a little about yourself?”. (line 220)

Comment 7

Lines 443-450 could be removed, you should imply what you need to do with the project from the results of this study.

Authors' response to comment 7:

Thank you for pointing this out. We have removed it to line 488-491, were we think it relates better to the findings of the present study.

Comment 8

Line 461-478 These data should be included in the results. 

Authors' response to comment 8:

Thank you for this important comment. We have added the findings in the result section, line 395-398:

“Overall, the possibility of answering online was perceived as acceptable and advantageously and The informants explained that because the questionnaire was online, they perceived it as an unintimidating approach. that gave them time and privacy to complete the questionnaires when the time felt right" and

And line 409-411:

“In contrast, some informants highlighted the advantages of an in-person approach, and one argued that continuity would be strengthened if the screening was conducted by one’s own midwife”.

Comment 9:

You should suggest your results for other countries, especially in LMIC could apply PHQ-9 as it is the regular screening for perinatal depression, such as India (R. Ransing, S.N. Deshpande, S.R. Shete, I. Patil, P. Kukreti, P. Raghuveer, et al.Assessing antenatal depression in primary care with the PHQ-2 and PHQ-9: can it be carried out by auxiliary nurse midwife (ANM)? Asian Journal of Psychiatry, 53 (2020), Article 102109) and Thailand. (Sawaddisan, R., Ransing, R., & Jatchavala, C. (2023). Concordance of the Thai versions of the Patient Health Questionnaire and Edinburgh Post-natal Depression Scale for antenatal depression. Journal of Health Science and Medical Research, 41(6), e2023985.), or recommend other screening tools (Srisurapanont, M.; Oon-arom, A.; Suradom, C.; Luewan, S.; Kawilapat, S. Convergent Validity of the Edinburgh Postnatal Depression Scale and the Patient Health Questionnaire (PHQ-9) in Pregnant and Postpartum Women: Their Construct Correlations with Functional Disability. Healthcare 2023, 11, 699.).

Authors' response to comment 9:

Thank you for this important suggestion.

We have added the following sentence in line 560-562:

“The present study was conducted in a western, high-income country. Future research should focus on examining the feasibility and experience of the screening model in low-income countries”.